# Thermal Preparation and Application of a Novel Silicon Fertilizer Using Talc and Calcium Carbonate as Starting Materials

**DOI:** 10.3390/molecules26154493

**Published:** 2021-07-26

**Authors:** Yian Wang, Jie Zhang, Junjian Zheng, Hua Lin, Gongning Chen, Chao Wang, Kong Chhuon, Zhonghua Wei, Chengfenghe Jin, Xuehong Zhang

**Affiliations:** 1College of Environmental Science and Engineering, Guilin University of Technology, 319 Yanshan Street, Guilin 541006, China; nickowya@163.com (Y.W.); WangChao06072021@163.com (C.W.); zhangxuehong@x263.net (X.Z.); 2College of Life and Environmental Science, Guilin University of Electronic Technology, 1 Jinji Road, Guilin 541004, China; 2012zhangjie@tongji.edu.cn (J.Z.); zhengjunjianglut@163.com (J.Z.); 3School of Chemistry and Materials Engineering, Huizhou University, 46 Yanda Road, Huizhou 516007, China; 4Faculty of Hydrology and Water Resources Engineering, Institute of Technology of Cambodia, Phnom Penh 12000, Cambodia; chhuon.k@gmail.com; 5Guilin Xinzhu Natural Functional Material Co., Ltd., Pioneer Park for Overseas Students, Guilin National High-tech Industrial Development Zone, Guilin 541004, China; info@da.com.cn (Z.W.); kaneshiro1688@yahoo.co.jp (C.J.)

**Keywords:** available Si, talc, CaCO_3_, calcination method, Si fertilizer

## Abstract

The deficiency of available silicon (Si) incurred by year-round agricultural and horticultural practices highlights the significance of Si fertilization for soil replenishment. This study focuses on a novel and economical route for the synthesis of Si fertilizer via the calcination method using talc and calcium carbonate (CaCO_3_) as starting materials. The molar ratio of talc to CaCO_3_ of 1:2.0, calcination temperature of 1150 °C and calcination time of 120 min were identified as the optimal conditions to maximize the available Si content of the prepared Si fertilizer. X-ray diffraction (XRD) and Fourier-transform infrared spectroscopy (FTIR) characterizations elucidate the principles of the calcination temperature-dependent microstructure evolution of Si fertilizers, and the akermanite Ca_2_Mg(Si_2_O_7_) and merwinite Ca_3_Mg(SiO_4_)_2_ were identified as the primary silicates products. The results of release and solubility experiments suggest the content of available metallic element and slow-release property of the Si fertilizer obtained at the optimum preparation condition (Si-OPC). The surface morphology and properties of Si-OPC were illuminated by the results of scanning electron microscope (SEM), surface area and nitrogen adsorption analysis. The acceleration action of CaCO_3_ in the decomposition process of talc was demonstrated by the thermogravimetry-differential scanning calorimetry (TG-DSC) test. The pot experiment corroborates that 5 g kg^−1^ soil Si-OPC application sufficed to facilitate the pakchoi growth by providing nutrient elements. This evidence indicates the prepared Si fertilizer as a promising candidate for Si-deficient soil replenishment.

## 1. Introduction

Silicon (Si), the second most abundant element in both the crust of earth and soil, has been broadly acknowledged as a quasi-essential nutrient for the growth of a variety of higher plants, particularly for crops such as rice, tomato, barley, sugarcane and soybean, which are crucial to human survival [1,2,3]. The essentiality of Si lies in its qualitative and quantitative promotion of higher plants, as it ameliorates soil conditions, increases nutrient contents in plants and enhances the resistance of plants to biotic (e.g., pathogen and insect or pest diseases) and abiotic (e.g., salinity, drought and ultraviolet radiation) stresses [1,4]. It is noteworthy that despite its ubiquity and abundance, the majority of Si appears in soil in the insoluble form, namely crystalline aluminosilicates that are unavailable for plants [2,5]. The scant presence of monosilicic acid (H_4_SiO_4_), known as the soluble and exclusive form of Si in soil phase that can permeate the root plasma membrane for plant utilization, has become a critical factor limiting the crop quality and yield in many regions worldwide, especially tropical and subtropical areas, where the soils are highly weathered [4,6,7]. It was estimated that the plant-available Si in paddy soil could be exhausted after consecutive cultivation for 5 years, provided that the crop growth was relied entirely on naturally occurring H_4_SiO_4_ [8].

The need for the elevation of crops yield justifies research aiming at developing efficient Si fertilizers to replenish the soils where a paucity of plant-available Si exists, utilizing extracts of natural minerals, chemical products and industrial wastes as starting materials. The results of field application have corroborated that Si fertilizers are of great agricultural and horticultural significance, attributed to their superior growth-promoting efficacy for a wide range of crops [9,10]. Relevant statistics show that, as a result of the field application of Si fertilizer, a 10 to 50% increment in sugarcane yield was attained in varying countries in Asia [2]. Nonetheless, the inherent drawbacks of the previously developed Si fertilizers, in terms of starting materials or the preparation process, severely limit their large-scale field application. For instance, as a quintessential natural mineral, wollastonite contains abundant Si (mainly CaSiO_3_) but suffers the labor-intensive refining process [1,11]; the use of soluble silicates such as Na_2_SiO_3_ and K_2_SiO_3_ to synthesize Si fertilizer is not economically feasible [12]; slag-based Si fertilizers involving carbon–steel slags, blast furnace slags, silicomanganese slags, phosphorus slags, etc. are relatively cost-effective alternatives; nevertheless, they raise environmental concerns because they potentially usher heavy metals into the soil [13]. In this context, it is high time that an efficient, inexpensive and eco-friendly Si fertilizer is developed.

Talc, a natural mineral with the chemical formula Mg_3_Si_4_O_10_(OH)_2_, has a three-layer structure composed of a magnesium–oxygen/hydroxyl octahedra layer sandwiched between two layers of silicon oxygen tetrahedra [14]. It is extensively employed as a dusting, coating and filler agent in paints, plastics, pharmaceuticals, rubber, lubricants, cosmetics as well as ceramics manufacture, owing to the well-documented merits involving great natural abundance, low-cost, high specific surface area, good physicochemical stability, etc. [15,16,17,18]. Moreover, recent evidence has proven its versatility as an efficacious absorbent for the elimination of aquatic toxic organic pollutants and heavy metals [18,19,20,21]. Given the foregoing advantages, as well as the fact that talc is known to be Si-rich (Si content of 63.3% in theory) [16] and contains hardly any poisonous heavy metals, it can be therefore extrapolated that talc holds the potential to be utilized as the raw material for Si fertilizer preparation. Notwithstanding, no single study to date covers the prospect of talc in application to Si fertilizer fabrication.

In this research, a novel Si fertilizer was prepared by the calcination method using talc and calcium carbonate as starting materials. In particular, the reason why calcium carbonate was incorporated into the preparation process is chiefly due to its functions in terms of lowering the melting temperature as well as facilitating the formation of calcium silicate [22]. The primary objectives of this study are to (1) evaluate how critical preparation conditions (i.e., the molar ratio of starting materials, calcination temperature and reaction time) correlate with effective Si content in synthetic fertilizers; (2) figure out the formation mechanism, surface morphology and physicochemical characteristics of prepared Si fertilizers; (3) disclose to what extent Si fertilizer application impacts the soil environment and growth of a target crop (i.e., pak choi).

## 2. Results and Discussion

### 2.1. Effects of Process Conditions on Available Si Contents of Prepared Fertilizers

Figure 1a–c delineate the variation of available Si content of the prepared Si fertilizers with a changing molar ratio of talc powder to CaCO_3_, calcination temperature and time. As can be seen from Figure 1a, regardless of the calcination temperature (1000 or 1100 °C), the available Si content was increased with the increasing molar ratio of talc powder to CaCO_3_ from 1:1.4 to 1:2.0 but decreased following the further augment of the proportion of CaCO_3_. This is unsurprising, because the appropriate addition of CaCO_3_ accelerated (i) the breakdown of talc powder, in view of the fact that the decomposition reaction (Equation (1)) of CaCO_3_ occurred along with massive heat release, and (ii) the reaction between the decomposition product (i.e., CaO) and talc powder to form available mineral silicates, while excessive CaCO_3_ dosage led to the decline in the mass percentage of available Si in the final products due to dilution effect [23]. Figure 1b exhibits that when the molar ratio of starting materials and calcination time was kept at 1:2.0 and 60 min, respectively, the available Si content was markedly increased from 6.5% at 1000 °C to 17.8% at 1150 °C, and a higher calcination temperature (1200 °C) resulted in an insignificant increase in the available Si content (*p* > 0.05). It can be inferred that a calcination temperature of 1150 °C was sufficient to allow the almost complete breakdown of talc power, which is distinctly lower than the reported calcination temperature (1300 °C) for accomplishing the transformation of pure talc [24]. This result suggests that the introduction of CaCO_3_ in the preparation process of Si fertilizer is conductive to reducing energy costs.
(1)$$CaCO_{3}\to CaO+CO_{2}$$

Figure 1c shows that in the case of the permanent molar ratio of starting materials (1:1.2) and calcination temperature (1150 °C), the extension of calcination time from 10 to 120 min gave rise to the increase in available Si content from 14.6% to 19.1%, implying that a proper calcination time can enable the complete decomposition of talc powder under the activation effect of CaCO_3_, followed by its effective conversion towards mineral fertilizers. Note that once the calcination time was beyond 150 min, the available Si content began to decrease. This is likely due to the fact that (i) diverse mineral silicates affiliated to different crystalline phases could be produced during the calcination process [25,26], and with the prolonging of the calcination time, those with poor thermodynamic stability were prone to transform into more stable ones which are resistant to leaching reagents, and/or (ii) the decomposition of a few mineral silicates into amorphous SiO_2_, as had been observed in the preparation process of Si fertilizer performed by Hu et al. [12]. Taken together, the molar ratio of talc powder to CaCO_3_ of 1:2.0, the calcination temperature of 1150 °C and calcination time of 120 min were chosen as the optimal preparation conditions, since they led to the relatively high available Si content in the final product while decreasing the energy costs to a certain extent.

### 2.2. Calcination Temperature-Dependent Microstructure Evolution of Si Fertilizers

According to the analysis results of Figure 1b as well as existing references [12,25,27], it is evident that calcination temperature is a decision factor affecting the composition and available Si content of synthetic Si fertilizers in sintering process. The XRD patterns of talc powder and Si fertilizers synthesized at varying calcination temperatures and/or times are depicted in Figure 2. Figure 2a shows the major diffraction peaks and intensities coincident with the structure characteristics of pure talc, according to the PDF 19-0770 powder diffraction file. After 1 h of calcination at 950 °C, the formation of kilchoanite Ca_6_(SiO_4_)(Si_3_O_10_) (PDF 46-1479) and clinoenstatite MgSiO_3_ (PDF 35-0610), as presented in Figure 2b, was ascribed to the reaction between the talc powder and CaO (Equation (2)). As shown in Figure 2c, a calcination temperature of 1000 °C resulted in the appearance of diopside CaMg(Si_2_O_6_) (PDF 82-0599) and monticellite CaMgSiO_4_ (PDF 84-1325), presumably associated with the interaction of the above primary decomposition products of talc powder and CaO (Equation (3)). Figure 2d exhibits that at 1050 °C, the talc powder was entirely exhausted, and the new crystalline phases of the final product involve larnite Ca_2_(SiO_4_) (PDF 86-0398), akermanite Ca_2_Mg(Si_2_O_7_) (PDF 83-1815), merwinite Ca_3_Mg(SiO_4_)_2_ (PDF 74-0382) and MgO (PDF 87-0651). This is likely attributed to either the further conversion reaction of kilchoanite, diopside and monticellite via Equations (4)–(7), or the addition reaction of intermediate products via Equation (8) [28]. In combination of parts e and f of Figure 2, it can be inferred that only two crystalline phases, i.e., akermanite and merwinite, were detected in the final product sintered at 1150 °C for 1 or 2 h; increasing calcination time from 1 to 2 h gave rise to the decrease in the intensity of the merwinite diffraction peak, along with the increase in intensity of akermanite diffraction peak, perhaps as a result of the transformation reaction of merwinite towards akermanite via Equation (9) [26,29].
(2)$$4\overset{talc\;powder}{Mg_{3}Si_{4}O_{10}{\left(OH\right)}_{2}}+6\overset{}{CaO}\to \overset{kilchoanite}{Ca_{6}(SiO_{4})(Si_{3}O_{10})}+\overset{clinoenstatite}{12MgSiO_{3}}+4H_{2}O$$
(3)$$\overset{kilchoanite}{Ca_{6}(SiO_{4})(Si_{3}O_{10})}+\overset{clinoenstatite}{7MgSiO_{3}}+CaO\to \overset{diopside}{4CaMg(Si_{2}O_{6})}+3\;\overset{monticellite}{CaMgSiO_{4}}$$
(4)$$\overset{kilchoanite}{Ca_{6}(SiO_{4})(Si_{3}O_{10})}+2CaO\to \overset{larnite}{4Ca_{2}(SiO_{4})}$$
(5)$$2\overset{diopside}{CaMg(Si_{2}O_{6})}+3CaO\to \overset{akermanite}{Ca_{2}Mg(Si_{2}O_{7})}+{\overset{merwinite}{Ca_{3}Mg(SiO_{4})}}_{2}$$
(6)$$\overset{diopside}{CaMg(Si_{2}O_{6})}+CaO\to \overset{akermanite}{Ca_{2}Mg(Si_{2}O_{7})}$$
(7)$$\overset{monticellite}{CaMgSiO_{4}}+CaO\to \overset{larnite}{Ca_{2}(SiO_{4})}+MgO$$
(8)$$\overset{larnite}{2Ca_{2}(SiO_{4})}+MgO+\overset{diopside}{CaMg(Si_{2}O_{6})}+CaO\to 2{\overset{merwinite}{Ca_{3}Mg(SiO_{4})}}_{2}$$
(9)$${\overset{merwinite}{Ca_{3}Mg(SiO_{4})}}_{2}\to \overset{akermanite}{Ca_{2}Mg(Si_{2}O_{7})}+CaO$$

Figure 3a1–f present the FTIR spectra of the CaCO_3_, talc powder and prepared Si fertilizers. The bands at 712, 877, 1430, 1800 and 2510 cm^−1^ can be attributed to the bending and asymmetric/symmetric stretching vibrations of C−O and C=O bonds of CaCO_3_ [30,31] (Figure 3a1). The band at 460, 617 and 671 cm^−1^ is associated with the Si−O−Si bending vibration [24,32], O−Si−O bending vibration and Si−O−Mg symmetric stretching vibration [12,24] of pure talc, respectively (Figure 3a2). As shown in Figure 3b–f, in the prepared fertilizers, the disappearance of the characteristic peaks of CaCO_3_ can be ascribed to the complete breakdown of CaCO_3_ in the calcination processes, since a temperature of 735 °C was known to enable the full decomposition of CaCO_3_ [33]; the absorption peak at 478 cm^−1^, relative to Si−O rocking vibration in the fully polymerized 3-D network [34], is presumably attributed to the generation of mineral silicates with diverse crystalline phases after calcination. This is also supported by observation of the new peaks at the 893–998 cm^−1^ region, which are pertaining to the stretching vibrations of mineral silicates containing functional groups with varying numbers of bridging oxygen atoms, e.g., [Si_2_O_6_]^4−^, [Si_2_O_7_]^6−^ and [Si_3_O_10_]^8−^ [26,34,35]. By comparing Figure 3(a2, b–f), it is evident that the Si−O−Si asymmetric stretching vibration was markedly shifted from 1018 cm^−1^ in talc to 1107 cm^−1^ in prepared fertilizers, presumably as a result of the occurrence of the dehydration of talc and the production of mineral silicates after thermal treatment [24,36]. The CO_3_^2−^ group-related band (corresponding to the adsorption peak at 1430–1515 cm^−1^ region [35]) was not found in prepared fertilizers, as shown in Figure 3b–f, implying the complete decomposition of CaCO_3_ at diverse calcination temperatures. The band at 1645 and 3400 cm^−1^ is assigned to the OH- bending and stretching vibration of absorbed water on all the tested samples, respectively, possibly arising from the moist laboratory air [24,33,35]. As displayed in parts e and f of Figure 3, the absence of characteristic Mg−OH stretching vibration of talc at 3678 cm^−1^ [37] indicates that the dehydroxylation of talc was not completed until heating to 1050 °C. The occurrence of some minor low-intensity peaks in parts e and f of Figure 3 is pertinent to the presence of SiO_2_ (PDF 14-0654) and Fe_2_O_3_ (PDF 39-1346), and the former was likely linked to the decomposition of mineral silicates [12], while the latter was perhaps due to the oxidization of Fe^2+^ in talc in the sintering processes, which is discussed below.

### 2.3. Dissolution Property and Physicochemical Characterization of Si-OPC Fertilizer

#### 2.3.1. Available Metallic Element Content and Solubility

Calcium and magnesium are essential elements for crops, as they normally play a central role in boosting growth and production as well as regulating defense mechanisms of crops against environmental stresses, but the intense foraging and harvesting of crops frequently leads to their depletion in large-scale soils [38,39]. Figure 4a indicates that after citric acid treatment, the released available content of calcium and magnesium in the Si-OPC was markedly larger than that in the talc powder (*p* < 0.05), attributed to the transformation of their metallic form from oxides to silicates; 15.35% and 5.69% of calcium and magnesium were released from the Si-OPC, which are relatively close to those (0.5–23% for calcium [12,27,40] and 0.2–4.4% for magnesium [25,27,41], respectively) in the reported Si fertilizers. As can be seen from parts a and b of Figure 4, similarly to talc powder, the Si-OPC exhibited poor available calcium and magnesium release capacity (<0.5%) and low solubility (0.0145 g/100 mL H_2_O) in the ultrapure water, implying the desirable slow-release property of the Si-OPC. In contrast, Na_2_SiO_3_, a typical quick-acting Si fertilizer with a high solubility (22.2 g/100 mL H_2_O), was susceptible to leaching from the root zone of crops by rainfall and potentially resulted in groundwater pollution [12,27]. Notably, the presence of Si-OPC in the ultrapure water led to the alkalization of solution, likely associated with the hydrolysis of merwinite (Equation (10)) and akermanite (Equation (11)). In this regard, the Si-OPC is particularly employed for nutrients replenishment in acidic soils, as it provides available Si and metallic elements while regulating soil pH.
(10)$$SiO_{4}^{4-}+4H_{2}O⇌H_{4}SiO_{4}+4OH^{-}$$
(11)$$Si_{2}O_{7}^{6-}+7H_{2}O⇌2H_{4}SiO_{4}+6OH^{-}$$

#### 2.3.2. Surface Morphology and Properties

The SEM images of talc powder are delineated in Figure 5a,b, suggesting the layered structure as well as smooth, plate-like and folding morphology of talc aggregates [19]. The irregular coarse and blocky (or granular) morphology of Si-OPC, as shown in Figure 5c,d, was likely due to the fact that the calcination with CaCO_3_ at a high temperature led to the destruction of the crystal lattice of talc power, followed by the reaction of decomposition products to generate silicates. The average grain diameter of Si-OPC was roughly 4.36 μm, in the range (0.8–230 μm) of reported Si fertilizers in the literature [12,40]. BET surface area analysis results indicate that the surface area of talc powder was 6.57 m^2^/g, significantly greater than that (1.33 m^2^/g) of Si-OPC. The reduction in surface area after calcination was presumably associated with the decrease in external pores (between the particles) and interparticle pores, as a result of the coarsening and densification of sintered particles [42,43]. According to the measurement results of pore size, the average pore size of Si-OPC was 18.81 nm, larger than that (14.18 nm) of talc powder, perhaps associated with the volume and/or grain boundary variations of particles at high temperature (>750 °C) [43]. On the grounds of adsorption isotherms of talc powder and Si-OPC (Figure 6), it is evident that the adsorption saturation phenomenon was not observed, implying the appearance of capillary condensation agglomeration in the nitrogen adsorption process [44]. The adsorption curves and desorption curves of talc powder and Si-OPC did not coincide, and the hysteresis loops were formed in the case of high relative pressure. With such a distinctive feature of Type H_3_ loop, the predominant pores of Si-OPC were speculated to be typical “slit type” mesopores [45].

#### 2.3.3. Thermogravimetry–Differential Scanning Calorimetry (TG-DSC) Analysis

The DSC curves of talc powder and starting materials (i.e., the mixture of talc powder and CaCO_3_) in Figure 7 manifest that weight increase was observed in the tested samples during the initial heating-up period, possibly attributed to the oxidation of Fe^2+^ and/or other unknown ingredients [46]. This is supported by our quantification result concerning the Fe ions content in talc, revealing that the Fe^2+^ content was 0.5718 ± 3.31% mg/g, accounting for 99.4% of the overall Fe ions (0.5754 ± 13.30% mg/g) in talc. The finding implies the possibility that there existed the oxidization of Fe^2+^ in talc towards Fe_2_O_3_ in the sintering processes. The apparent weight loss in the DSC curve of talc powder (Figure 7a), occurring over the temperature range from 800 to 1000 °C, was assigned to the generation of enstatite, amorphous silica and water, as a consequence of the dehydroxylation of talc powder [24]. In the DSC curve of the starting materials (Figure 7b), the significant weight loss at 600–734 °C was predominantly associated with the breakdown of CaCO_3_. Analogously, Armina et al. found that the thermal decomposition of CaCO_3_ could proceed until 735 °C [33]. The weak exothermic peak centered at 734 °C was perhaps due to the reaction between CaO and water (generated by the dehydroxylation of talc powder). Another obvious weight loss after 860 °C was likely correlated to the CaO-driven decomposition reaction of talc powder (Equation (2)). According to the literature, when pure talc was used in the sintering process, the breakdown temperature of talc was in the temperature range of 800–1300 °C, and the clinoenstatite, an intermediate product of talc, began to generate at 1200 °C [24]. However, in this research, a calcination temperature of 950 °C was sufficient to produce clinoenstatite (Figure 2b). Hence, the presence of CaCO_3_ is beneficial for accelerating the conversion process of talc to silicates. This is in line with the findings obtained by Rashita et al., who reported that the introduction of CaCO_3_ in sintering process led to the low-temperature production of wollastonite [47].

### 2.4. Effects of Silicon Fertilizer on Soil Environment and Growth of Pak Choi

As depicted in Figure 8a, in comparison to the control treatment, 5–60 g kg^−1^ soil Si-OPC dosage led to the increase in exchangeable magnesium, calcium and Si content in soil from 0.72, 7.77 and 3.87 cmol kg^−1^ soil to 1.55–5.58, 9.47–12.19 and 6.45–19.10 cmol kg^−1^ soil, respectively. In contrast to the variation tendency of Si content in soil, the relatively slower increasing trend of magnesium, calcium content with the increase in Si-OPC application can be ascribed to their co-precipitation with carbonate, phosphate and hydroxyl ions in soil solution. Soil pH was proven to be significantly correlated to the uptake capacity of pak choi for macronutrients (e.g., available N and P), and a pH of 6.0–7.0 was the optimal condition for pak choi growth [48,49]. The determination results of soil pH manifest that once the dosage of Si-OPC was beyond 20 g kg^−1^ soil, the resultant alkaline environment might be deleterious to pak choi growth. Notably, the augment of soil pH with increasing application of Si-OPC was attributed to the hydrolysis of merwinite (Equation (10)) and akermanite (Equation (11)). Parts c, d and e of Figure 8 exhibit the correlation between the growth indexes of pak choi and the dosage of Si-OPC. Obviously, 5 g kg^−1^ soil Si-OPC application amount was mostly effective for pak choi growth; for instance, it contributed to a 15.63–295.35% significant increase (*p* < 0.05), in terms of plant height, germination rate, fresh weight, plant height and root length of pak choi, compared to those of pak choi without Si-OPC application. Additionally, a dosage of 5 g kg^−1^ soil Si-OPC resulted in an available Si content of 11.79 mg g^−1^, markedly higher than that (6.44 mg g^−1^) of the control group (Figure 8f). These findings are in accordance with previous results that Si fertilization was capable of significantly increasing the yields of crops such as rice, cucumber, tomato and pak choi [48,50]. Hu et al. found that, compared to the control group, an analogous Si dosage (4 g kg^−1^ soil) was sufficient to increase the germination rate, plant height, root weight and fresh weight of pak choi [12]. With regard to the declined growth indexes and available Si content in pak choi, as Si-OPC dosage was increased from 5 to 60 g kg^−1^ soil, in addition to the soil pH rise, another possible reason accounting for this phenomenon is the increase in soil salinity, which might give rise to (1) the inhibition of photosynthesis, owing to the triggered close of stomata and disturbance of the CO_2_-to-O_2_ ratio in leaves [51], (2) and the plasmolysis of plant cell because of the exorbitant osmotic pressure, and therefore impair the cellular metabolism [6]. In particular, in the case of the optimum Si-OPC application (5 g kg^−1^ soil), the resultant exchangeable calcium content in the soil was 9.47 cmol/kg soil, with this value in the range of moderate calcium level (5–10 cmol/kg) in soil [52]. Despite the fact that excessive calcium dosage can hamper the growth and nutrients uptake of plants, owing to the induced ions imbalance in plants [53], a prior reference suggested that a similar calcium concentration in soil (10.7 cmol/kg soil) had no adverse impact on plant growth [54].

## 3. Materials and Methods

### 3.1. Materials

Talc with a purity of 97.3% was provided by Guangxi Longguang Talc Development Co., Ltd., Guilin, China. The chemical composition of talc was determined by ZSX Primus II X-ray fluorescence spectrometry (XRF, Rigaku Corporation, Tokyo, Japan) with the results showing that it consists of Al_2_O_3_ 0.40 wt%, CaO 0.13 wt%, Fe_2_O_3_ 0.54 wt%, MgO 31.04 wt%, Na_2_O 0.08 wt%, P_2_O_5_ 0.03 wt%, SiO_2_ 61.01 wt% and LOI 6.67 wt%. Prior to thermal treatment, the talc was crushed and ground to powder by a ball mill (QM-3SP2, Nanjing Keyscience Electronic Technology Co., Ltd., Nanjing, China), and passed through a 75 μm sieve (200 mesh). All the chemicals such as calcium carbonate (CaCO_3_) and citric acid monohydrate were of analytical grade, purchased from Sinopharm Group Chemical Reagent Co., Ltd., Shanghai, China. The obtained talc powder and CaCO_3_ were dehydrated in an oven at 110 °C for 3 h to constant weight, and then placed in a desiccator for storage until use.

### 3.2. Preparation of Si Fertilizers

To reveal the optimum condition for Si fertilizer preparation, single-factor experiments were conducted for evaluation of the variation trends of available Si content in the prepared Si fertilizers, as a function of changing molar ratios between talc powder and CaCO_3_ (i.e., 1:1.4, 1:1.6, 1:1.8, 1:2.0, 1:2.2 and 1:2.4), sintering temperatures (i.e., 950, 1000, 1050, 1100, 1150 and 1200 °C) and reaction time (i.e., 10, 30, 60, 90, 120, 150, 180 and 240 min). In all experiments, talc powder and CaCO_3_ were evenly mixed, followed by the transfer of the mixture to a ceramic crucible, and the subsequent sintering in a muffle furnace (SRJX-4-13A, Zhejiang Shangyu Fashion Instrument, Shaoxing, China) operated with a heating power of 4 kW at air atmosphere. After sintering, the prepared Si fertilizers were cooled down naturally, and then milled to around 150 μm manually in an agate mortar prior to analysis and/or use.

### 3.3. Release Performance and Solubility Tests

To evaluate the available elements release capacity and solubility of Si-OPC (namely, the Si fertilizer obtained at the optimum preparation condition), experiments were carried out to examine the contents of available elements such as Si, calcium, magnesium and aluminum in liquid- and solid- phases, after the Si-OPC was treated by leaching reagents, i.e., 2% citric acid [55] and ultrapure water; meanwhile, the raw material, i.e., talc powder, was used as the control. Citric acid was employed as the leaching reagent mainly because of its massive secretion by the rhizosphere of plants [56], and the superior capacity to form a thermodynamically stable complex over other commonly adopted leaching reagents (e.g., hydrochloric acid) [57]. The detailed experimental processes are the following: 0.5 g of samples was weighed and mixed with 250 mL of leaching reagents; the mixtures were shaken for 30 min at 29 °C and 190 r/min in a thermostatic reciprocating oscillator (ZD-88, Jiangsu Dadi Automation Instrument Factory, Changzhou, China); after filtration, filtrates were taken out for pH and elemental contents analysis, while residues underwent mass loss analysis.

The available Si contents of filtrates were determined by a UV-Vis spectrophotometer (UV-6100, Shanghai Mapada Instruments Co., Ltd., Shanghai, China) using the silicon–molybdenum blue colorimetric method [58]. The measurements of available metallic elements in the filtrates were performed on an inductively coupled plasma optical emission spectrometer (ICP-OES, Optima 7000 DV, PerkinElmer, Wellesley, MA, USA), according to the analytical protocols (NY/T 2272-2012) released by the Chinese Ministry of Agriculture. The pH of filtrates was measured by a pH meter (PHS-3E, Shanghai INESA Scientific Instrument Co., Ltd., Shanghai, China). After leaching reagents treatment and filtration, the residues of Si-OPC and talc powder were placed in a crucible and calcined at 1000 °C in the muffle furnace for 30 min. The solubility of samples was deduced by the determination of the mass loss after calcination.

The experimental process for quantification of available Fe^2+^ and overall Fe contents in talc is as follows: 0.5 g of talc powder was weighed, and then suspended in 100 mL of aqueous solution (pH = 3); after that, the mixture was vibrated at 29 °C and 190 r/min for 24 h; subsequently, the suspension was taken out for the available Fe^2+^ and overall Fe contents analysis, according to the O-Phenanthroline spectrophotometric method (HGT 3539-1990) released by the Chinese Ministry of Chemical Industry.

### 3.4. Composition and Microstructural Characterization

The surface morphology of talc powder and Si-OPC was studied on an SU5000 scanning electron microscope (SEM, Hitachi Limited, Hitachi, Japan) operating at an acceleration voltage of 5 kV and an emission current of 110 μA. The pyrolysis characteristics of the starting materials were investigated using an STA 8000 thermogravimetric analyzer (TG-DTA, PerkinElmer, Wellesley, MA, USA) in air atmosphere (gas-flow rate = 20 mL/min) at the heating rate of 5 °C/min from 30 to 1300 °C. The chemical composition analysis of talc powder and prepared Si fertilizers was performed on a X’Pert PRO X-ray diffractometer (XRD, PANalytical, Almelo, Netherlands) at a scan step size of 0.02° in the 2θ range of 5 to 90° with a working voltage and current of 40 kV and 40 mA, respectively, using Cu Kα as a radiation source. The structural analysis of functional group of CaCO_3_, talc powder and prepared Si fertilizers was carried out by a Nicolet FT-IR 6700 Fourier-transform infrared spectrometer (FTIR, Thermo-Scientific, Waltham, MA, USA) in the wavenumber range of 400 to 4000 cm^−1^ at room temperature, using the KBr dispersion method. The Brunauer–Emmett–Teller specific surface area and pore size of talc powder and Si-OPC were evaluated by nitrogen adsorption at 77 K on a Gemini III-2375 full-automatic surface area analyzer (BET, Micromeritics, Atlanta, GA, USA).

### 3.5. Pot Experiments

“Siji” pak choi seeds received from Hezhiyuan Seed Industry were planted in cylindrical plastic pots (diameter = 20 cm, height = 19 cm). Before use, the pak choi seeds were sterilized by immersion in 10% (*v/v*) H_2_O_2_ solution for 15 min, rinsed thoroughly with distilled water and soaked in water for 24 h. Each pot was loaded with 1.5 kg of air-dried and sieved (0.85 mm) soil that was obtained from Guilin Yanshan Garden. Completely mixed basal fertilizers, including 0.35 g kg^−1^ soil of urea (CO(NH_2_)_2_), 0.31 g kg^−1^ soil of KH_2_PO_4_ and different dosages of Si-OPC (i.e., 0, 5, 10, 20, 40 and 60 g kg^−1^ soil) were supplied to every pot. These pots, in which 20 pak choi seeds were evenly planted, were placed in a greenhouse at a relative humidity of 60% and a day/night temperature of 30 °C (12 h): 25 °C (12 h) and watered every 3 days with 100 mL of distilled water throughout the experimental period. After 10 days of incubation, the germination rates were recorded; on day 40, the mature plants were harvested for the measurement of height/length, fresh weight, dry weight and the Si content of plants and/or roots, and the soils were collected for properties analysis.

The Si concentration of plants was determined according to the method identical to Yin et al. [2] Briefly, 0.05 g of the dried samples was weighed, milled and transferred to the crucible for calcination at 550 °C for 3 h; then, the remaining ashes were extracted by the mixture of 40% HF and 0.08 mM H_2_SO_4_, followed by detecting the Si concentration of the extracts. Soil pH was measured in 1:2.5 (*w*:*v*) soil/water suspensions [6,59]. Regarding the quantification of available metallic elements in soil, available Si was extracted by 0.025 mol/L citric acid, and the available calcium and magnesium were assayed after the soil samples were digested with HNO_3_-HF-HClO_4_ system [59].

### 3.6. Data Analysis

All data were processed by Microsoft Excel 2019 and Origin 2018, and the statistical analysis was conducted by SPSS 26.0 and Duncan’s method for significant difference analysis.

## 4. Conclusions

In this work, talc powder and CaCO_3_ were mixed at a molar ratio of 1:2.0 and calcined at 1150 °C for 120 min to attain the desirable Si fertilizer (i.e., Si-OPC) with an available Si content of 19.1%. The temperature-driven reactions involving the decomposition reaction of talc powder and CaCO_3_ as well as the recombination reaction of their intermediate products led to the generation of akermanite Ca_2_Mg(Si_2_O_7_) and merwinite Ca_3_Mg(SiO_4_)_2_ as the dominant final silicates products. The preferable nutrient element release amount and solubility of the Si-OPC in citric acid instead of ultrapure water suggests its potential as a slow-release fertilizer. The Si-OPC exhibits an irregular coarse and blocky morphology with an average pore size of 18.81 nm, and its nitrogen adsorption curves conform to a distinctive feature of Type H_3_ loop. CaCO_3_ functioned as the accelerator and participated in the conversion process of talc to mineral silicates. According to the results of the pot experiment, an application amount of 5 g kg^−1^ soil Si-OPC was mostly effective for pak choi growth, and a higher dosage resulted in the sharp decline in growth indexes of pak choi, presumably due to the soil pH rise and/or the increase in soil salinity. Therefore, on the premise of precise dosage control, Si-OPC might be a potential alternative for boosting crops production in available Si-deficient soil.

## Figures and Tables

**Figure 1 molecules-26-04493-f001:**
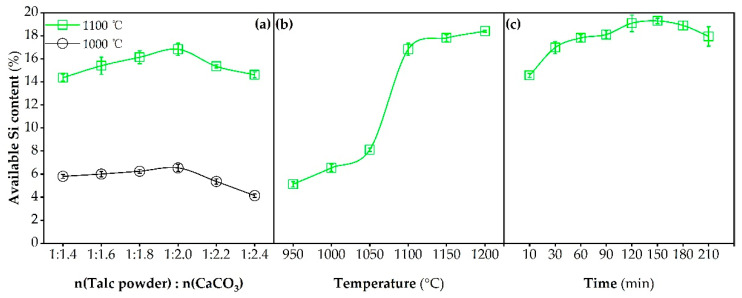
Effects of the (**a**) molar ratio of talc powder to CaCO_3_ (1:1.4–1:2.4), (**b**) calcination temperature (950–1200 °C) and (**c**) calcination time (10–210 min) on available Si contents of the prepared Si fertilizers. In plot (**a**), calcination time = 60 min; in plot (**b**), calcination time = 60 min, n(Talc powder): n(CaCO_3_) = 1:2.0; in plot (**c**), calcination temperature = 1150 °C, n(Talc powder): n(CaCO_3_) = 1.2.

**Figure 2 molecules-26-04493-f002:**
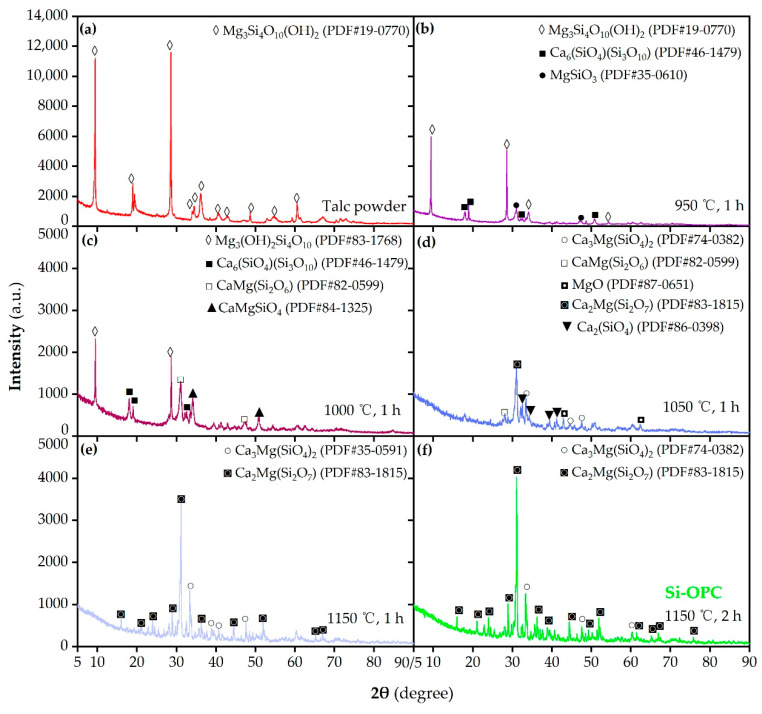
XRD patterns of the (**a**) talc powder and the prepared Si fertilizers after calcination at (**b**) 950 °C, (**c**) 1000 °C, (**d**) 1050 °C and (**e**) 1150 °C for 1 h, as well as (**f**) 1150 °C for 2 h (i.e., the Si-OPC). In all cases, n(Talc powder): n(CaCO_3_) = 1:2.0.

**Figure 3 molecules-26-04493-f003:**
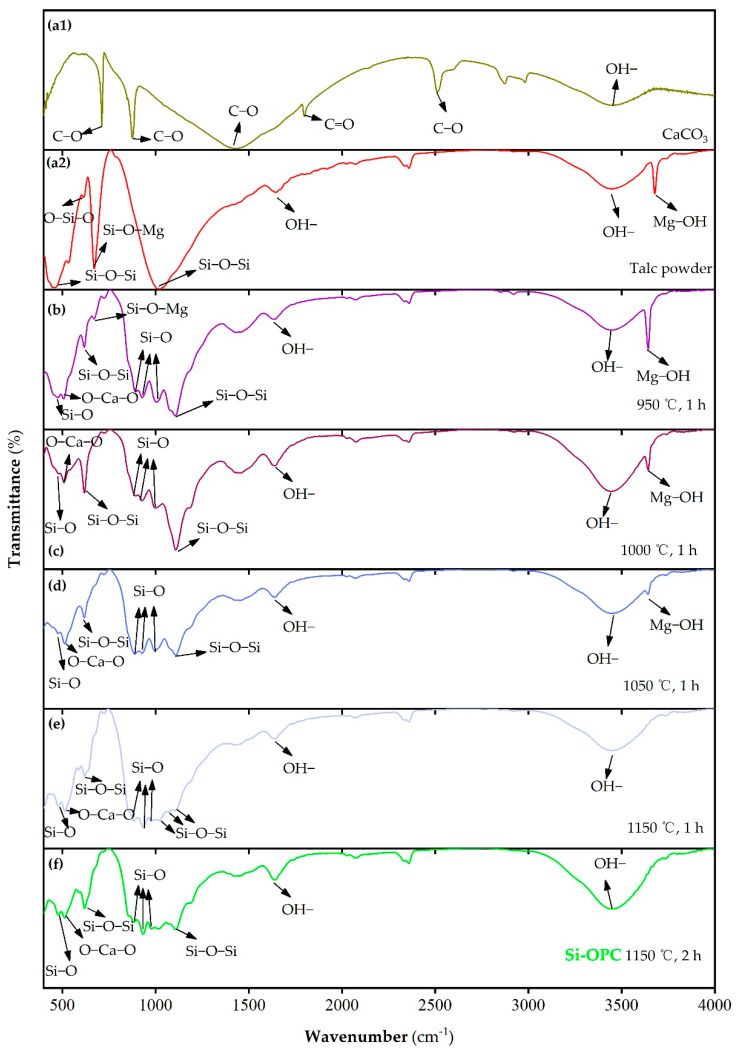
FTIR spectra of the (**a1**) CaCO_3_, (**a2**) talc powder and the prepared Si fertilizers after calcination at (**b**) 950 °C, (**c**) 1000 °C, (**d**) 1050 °C and (**e**) 1150 °C for 1 h, as well as (**f**) 1150 °C for 2 h (i.e., the Si-OPC). In all cases, n(Talc powder): n(CaCO_3_) = 1:2. All samples were prepared as KBr pellets.

**Figure 4 molecules-26-04493-f004:**
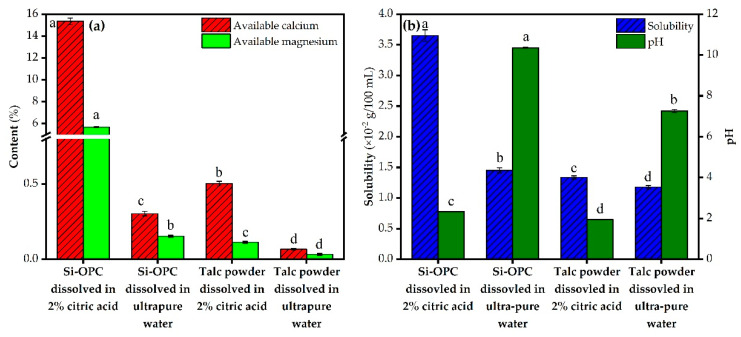
Comparison of (**a**) the available calcium, magnesium content and (**b**) the solubility of Si-OPC and talc powder as well as the pH of leaching solution when 2% critic acid and ultrapure water was used as leaching reagent, respectively. Means with different letters above the bars are statistically different from each other at *p* < 0.05.

**Figure 5 molecules-26-04493-f005:**
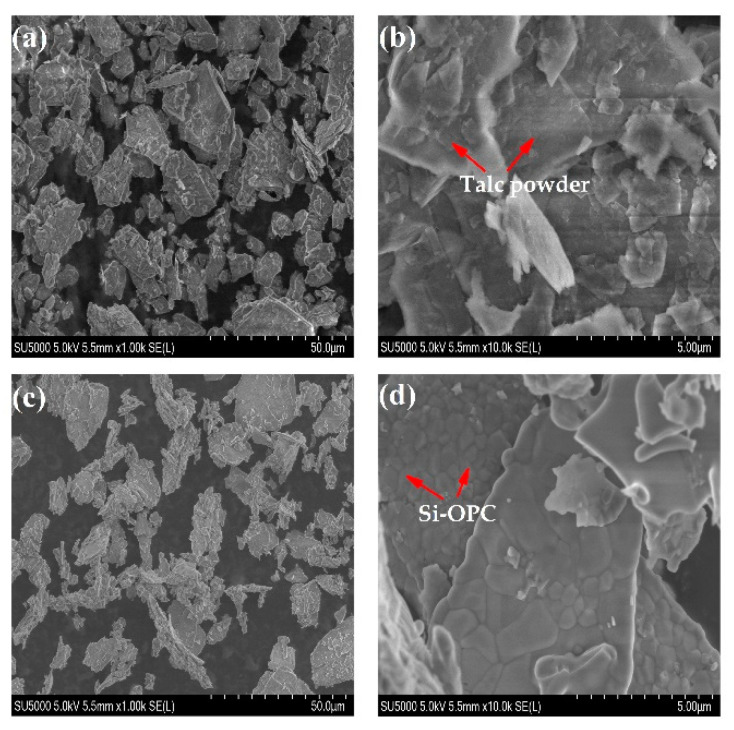
SEM images of talc powder ((**a**), 1.00 k; (**b**), 10.0 k) and Si-OPC fertilizer ((**c**), 1.00 k; (**d**), 10.0 k).

**Figure 6 molecules-26-04493-f006:**
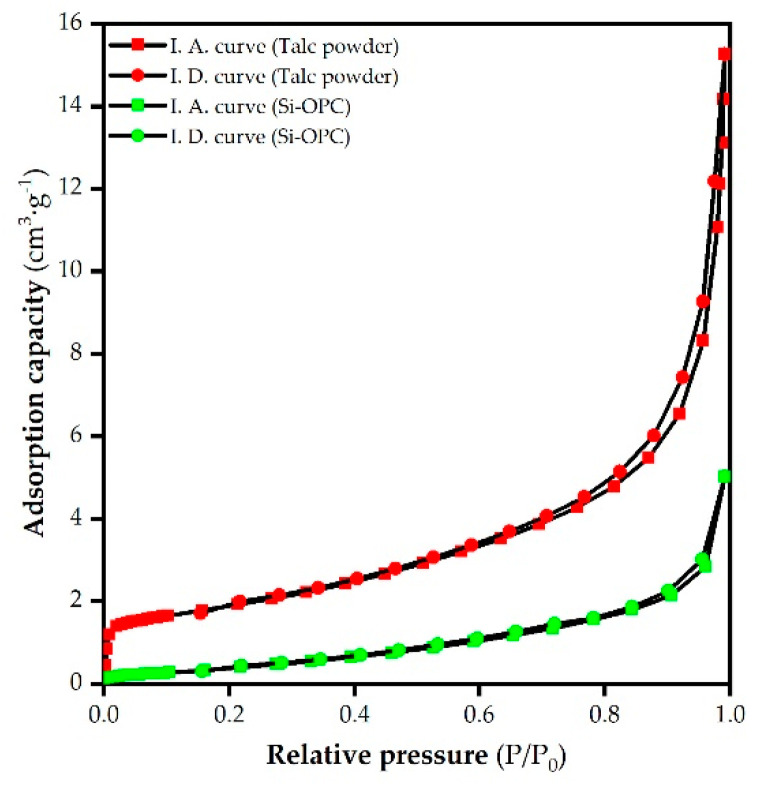
Isothermal adsorption (IA) and desorption (ID) curves of talc powder and Si-OPC fertilizer.

**Figure 7 molecules-26-04493-f007:**
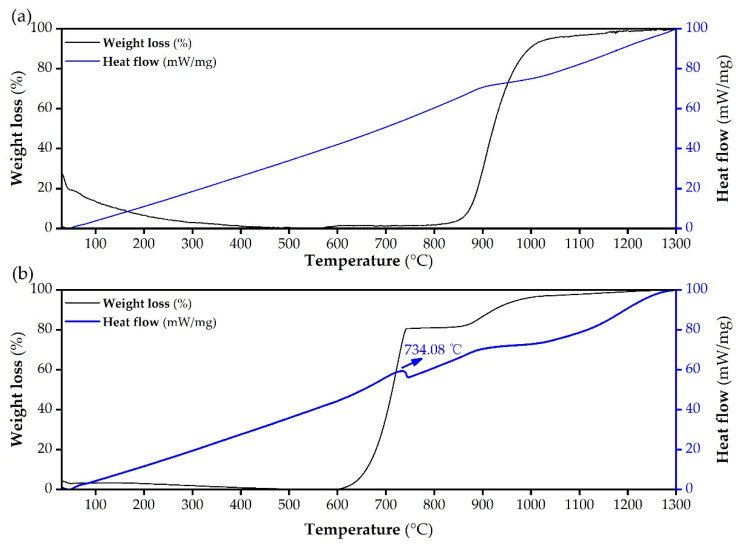
TG-DSC curves of (**a**) talc powder and (**b**) the mixture of talc powder and CaCO_3_.

**Figure 8 molecules-26-04493-f008:**
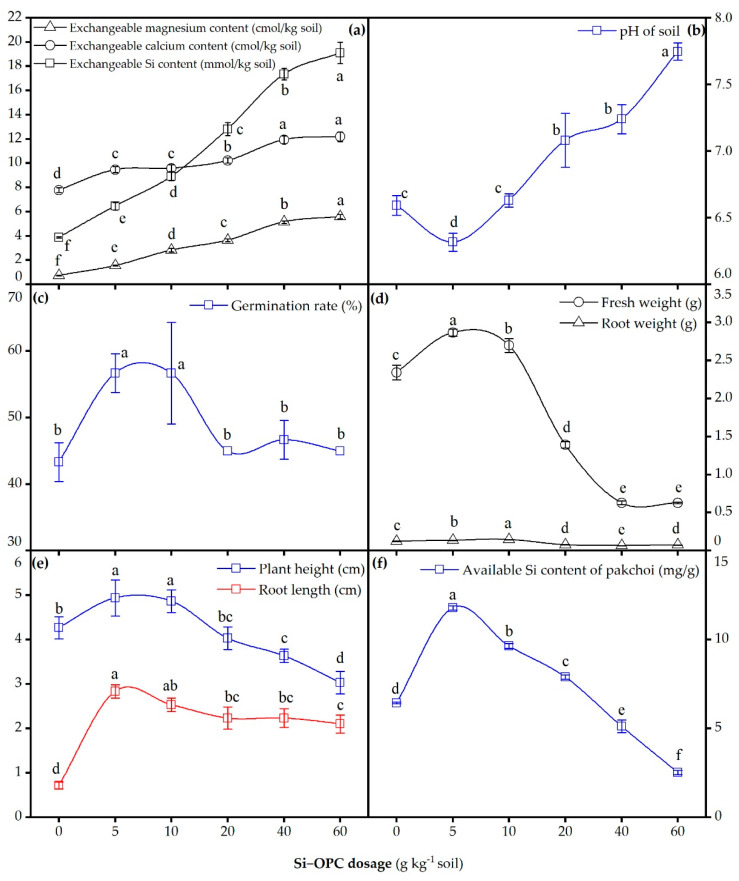
Effects of Si-OPC dosage (0–60 g kg^−1^ soil) on (**a**) exchangeable magnesium, calcium and Si content and (**b**) pH of soil as well as (**c**) germination rate, (**d**) fresh weight and root weight, (**e**) plant height and root length and (**f**) available Si content of pak choi. Means with different letters above the spots are statistically different from each other at *p* < 0.05.

## Data Availability

Data are contained within the article.

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
