# Peer review of "Thermal Preparation and Application of a Novel Silicon Fertilizer Using Talc and Calcium Carbonate as Starting Materials"

_molecules, 2021, doi:10.3390/molecules26154493_

Round 1
Reviewer 1 Report
How do the authors explain why the maximum of many plant parameters was attained at 5g Si-OPC/kg soil? Is there in literature any data on the effect of Si concentration on the studied plant parameters? Can the authors compare the effect of Si-OPC on plant parameters with the literary data for Si concentration dependence of the same plant parameters?
Statistical analysis is not explained, in the part related to the measurements of released elements’ content (Fig. 4), and measurements of plant parameters (Fig. 8). In Materials and methods section the authors should describe statistical data analysis. Also, in the figure legends it should be explained what show the error bars- standard error od deviation, and which was the significance level.
Page 11, line 316: ... (i.e., 1:1.4, 1:1.6, 1:1.8, 1.20, 1.22 and 1.24). What the numbers 1.2, 1.22 and 1.24 mean? It may be an error.
Author Response
Thanks for the incisive comment. Please see the attachment.

Reviewer 2 Report
Footnote of Figure 3 must be completed to add the procedure used in the FTIR measurements. That is the samples have been prepared as KBr pellets. This is important since the authors assigned broad peaks around 1600 and 3400 cm-1 to the presence of physisorbed water. I agree these peaks are due to water in KBr pellets. That is because the label OH- in the FTIR spectra must be changed by H2O.
For comparative purposes, the carbonate FTIR should also be included in figure 3.
FTIR samples show different peaks in the range attributed to Si-O-Si bonds. In addition, DRX obtained at high temperatures shows an important noise/signal ratio. Is it possible that the calcination provides amorphous silicate phases not detected by DRX?
The format of the TG-DSC curves is not easy to understand at first view. The curves should be normalized, and the weight loss represented from higher to lower values (%) (and the inverse for the weight gain). See for example reference 45.
The authors justify the small weight gain in tested samples during the initial heating-up period to the presence of Fe2+ and other unknown ingredients. The ICP or FRX measurements of these samples should be able to corroborate this point.
The tendency of Si content in soil is clearly dosage fertilizer dependent. On the contrary, the increasing tendency of Ca and Mg is slower. 5g/Kg dosage seems to offer better results in terms of plant growth, at this point Ca availability is even higher than Si, which is related to the presence of merwinite and akermanite phases very rich in this metal. Has the presence of a higher amount of Ca a direct effect on plant growth or is it strictly due to the presence of Si.
Author Response

(The authors gave the same response as above.)

Round 2
Reviewer 2 Report
After reading the revised manuscript I considered that is ready to be published in Molecules in its present format.
Congratulations.